# A Comparison among Nonvitamin K Antagonist Oral Anticoagulants in Asian Patients with Venous Thromboembolism: A Multi-Institutional Study

**DOI:** 10.3390/jcm11237159

**Published:** 2022-12-01

**Authors:** Ming-Lung Tsai, Cheng-Hung Lee, Ming-Jer Hsieh, Shao-Wei Chen, Shang-Hung Chang, Chi-Nan Tseng, Pao-Hsien Chu, I-Chang Hsieh, Po-Chuan Ko, Yu-Tung Huang, Dong-Yi Chen

**Affiliations:** 1Division of Cardiology, Department of Internal Medicine, New Taipei Municipal TuCheng Hospital, Taiwan, and Chang Gung University College of Medicine, Taoyuan 333, Taiwan; 2Division of Cardiology, Department of Internal Medicine, Chang Gung Memorial Hospital at Linkou and Chang Gung University College of Medicine, Taoyuan 333, Taiwan; 3Department of Thoracic and Cardiovascular Surgery, Chang Gung Memorial Hospital at Linkou and Chang Gung University College of Medicine, Taoyuan 333, Taiwan; 4Center for Big Data Analytics and Statistics, Department of Medical Research and Development, Chang Gung Memorial Hospital at Linkou, Taoyuan 333, Taiwan

**Keywords:** pulmonary embolism, deep vein thrombosis, anticoagulation, bleeding, venous thromboembolism

## Abstract

The comparison of clinical effectiveness and safety across different nonvitamin K antagonist direct oral anticoagulants (DOACs) in Asian patients with venous thromboembolism (VTE) remains unclear. Therefore, we assessed the real-world benefits of different DOACs in these patients. A cohort of 1480 patients with VTE were identified from the Chang Gung Research Database between 1 January 2012, and 31 December 2019. The composite outcomes of recurrent VTE and major bleeding were evaluated for four DOACs. The composite outcomes of recurrent VTE and major bleeding occurred in 9.06%, 9.80%, 8.61%, and 10.86% of the apixaban, dabigatran, edoxaban, and rivaroxaban groups, respectively, within 12 months of treatment initiation. The risk of the composite outcomes was similar in the rivaroxaban group and the apixaban, dabigatran, and edoxaban groups, with a subdistribution hazard ratio (SHR) of 0.80 (95% CI, 0.49–1.29), 0.81 (95% CI, 0.34–1.95), and 0.76 (95% CI, 0.42–1.39), respectively. No significant differences in the rates of recurrent VTE or major bleeding were observed between the rivaroxaban and other DOAC groups at the 12-month follow-up. According to real-world practice in Asian patients with VTE, the DOAC type was not associated with the differences in the risk of recurrent VTE or major bleeding within 12 months of treatment initiation.

## 1. Introduction

The incidence of venous thromboembolism (VTE), comprising deep vein thrombosis (DVT) and pulmonary embolism (PE), is approximately 1 in 1000 persons each year [1]. Death within one year occurs in approximately 14.6% of patients with DVT and 52.3% of those with PE without proper treatment [2]. In patients receiving treatment, recurrent DVT or PE may lead to a postthrombotic syndrome, chronic thromboembolic pulmonary hypertension, and death [3]. Nonvitamin K antagonist direct oral anticoagulants (DOACs) are used as the standard treatment for patients with VTE. Randomized controlled trials (RCTs) have demonstrated that the efficacy of DOACs for VTE is similar to that of the traditional warfarin treatment but with significantly lower major bleeding rates [4,5,6,7,8,9]. DOACs are preferred over warfarin because of lower major bleeding rates and a more reliable therapeutic range [10]. Most studies on VTE treatment have compared DOACs with warfarin; however, studies comparing the clinical benefits of various DOACs in the real world are limited. Data about comparisons of DOACs may improve the optimal care for patients with VTE. Furthermore, studies on the effectiveness of DOACs in Asian patients with VTE are limited. Although the results of pivotal trials in Asian patients have suggested that DOACs present a possible safety advantage over other treatments, real-world data on effectiveness and safety comparisons between DOACs in Asian patients with VTE are lacking [11]. To better understand the outcomes of various DOACs in Asian patients with VTE, we evaluated real-world evidence on the effectiveness and safety of DOACs in patients with VTE.

## 2. Methods

### 2.1. Data Sources

The Chang Gung Research Database (CGRD) was used for this study, which is a de-identified database maintained by Chang Gung Memorial Hospital (CGMH), a multi-institution system and the largest health care provider in Taiwan [12,13]. From the CGMH system, the CGRD contains data on emergency services, inpatients, and outpatients. Disease diagnoses and procedures were recorded using the *International Classification of Diseases, Ninth Revision, Clinical Modification* (*ICD-9-CM*) codes before 2016, whereas the *ICD-10-CM* codes have been used after 2016. The admission data, discharge status, and mortality coding in the CGRD have been validated, with the results revealing their high accuracy [14]. The *ICD-9-CM* and *ICD-10-CM* diagnosis codes used in this study are listed in Appendix A.

### 2.2. Study Population and Design

Patients with newly diagnosed VTE who were hospitalized and treated with DOACs between 1 January 2012 and 31 December 2019 were identified from the CGRD. This diagnosis of VTE included: (1) patients with a discharge diagnosis of DVT or PE using the *ICD-9-CM* or *ICD-10-CM* codes (Appendix A), (2) radiographically confirmed DVT (by using duplex ultrasonography) or PE (by using chest computed tomography (CT) scan or lung perfusion scan), and (3) an outpatient diagnosis of DVT or PE made at least twice and with the subsequent use of DOACs were included. To investigate the effect of only one DOAC, we excluded patients who were treated with another DOAC within 30 days of treatment initiation. The date of the first DOAC prescription was defined as the index date. After the index date, patients were followed up until the first occurrence of the outcomes, 1 year after the index date, death, or the end of the follow-up period (December 2019) whichever came first. Furthermore, to exclude prevalent cases and reduce detection bias, patients with new effectiveness events occurring within 5 days of the index date were excluded. The study cohort enrollment process is displayed in Figure 1.

### 2.3. Exposure and Covariates

The evaluated DOACs were rivaroxaban (ATC code B01AF01), apixaban (ATC code B01AF02), edoxaban (ATC code B01AF03), and dabigatran (ATC code B01AE07) with the dosage regimen being either standard-dose regimen or low-dose regimen. According to the Taiwan National Health Insurance regulation for VTE treatment, the standard-dose regimen for DOACs is 5 mg twice per day for apixaban, 60 mg once per day for edoxaban, 150 mg twice per day for dabigatran, and 20 mg once per day for rivaroxaban. The patients who received doses lower than the standard dose were classified as the low-dose group. The covariates included age; sex; body mass index (BMI); baseline levels of hemoglobin, platelet, creatinine, and D-dimer; Charlson comorbidity index; and comorbidities. Data on the use of medications, including aspirin, P2Y12 inhibitors, and nonsteroidal anti-inflammatory drugs, 30 days before DOAC initiation were extracted.

### 2.4. Outcomes

In the analysis, the involvement of a new blood vessel or new thrombus formation confirmed by chest CT scan, lung perfusion scan, or duplex ultrasonography was defined as the recurrent VTE. An intracranial hemorrhage, major gastrointestinal (GI) bleeding, other critical site bleedings, or a hemoglobin decrease of ≥2 g/dL over 24 h was considered a major bleeding event [15]. The primary outcomes were defined as the composite endpoints of effectiveness and safety. In addition, each effectiveness and safety outcome was measured. Data on safety outcomes were retrieved from the CGRD (Appendix A).

### 2.5. Statistical Analysis

Continuous variables are presented as mean ± standard deviation, and categorical data are presented as frequency (percentage). Differences between apixaban, dabigatran, edoxaban, and rivaroxaban users were tested using the analysis of variance for continuous variables and the χ^2^ test for categorical variables. We compared the risks of recurrent VTE (effectiveness outcome) or major bleeding (safety outcome) between the groups using Cox proportional hazards models with multivariable adjustments for all the covariates (Table 1). The risks of time-to-event outcomes in the groups were further compared using a Fine and Gray subdistribution hazard model that considered death a competing risk. The rivaroxaban group was used as the reference group because more than half of the patients used rivaroxaban. A subgroup analysis was performed to determine the hazard ratios (HRs) of the composite outcomes for different DOAC groups in the prespecified subgroups, including sex, age ≥ 65 years, pulmonary embolism, renal function (eGFR ≤ 50 mL/min/1.73 m^2^), hemoglobin level ≤ 10 g/dL, standard DOAC dose, low DOAC dose, and active cancer status. Finally, we created a stabilized inverse probability of treatment weighting (IPTW) cohort based on the propensity score, which was calculated using the values of the covariates listed in Appendix A. The risks of recurrent VTE or major bleeding between the groups were compared using a Fine and Gray subdistribution hazard model that considered death a competing risk. To further rule out possible residual confounding even after IPTW, we adjusted for the covariates with an absolute STD value > 0.1 in the survival models. All statistical analyses were performed using SAS 9.4 (SAS Institute, Cary, NC, USA). All statistical tests were 2-sided, and a *p* value of <0.05 was considered to be significant.

### 2.6. Ethics Statement

The CGMH Institutional Review Board approved this study (IRB No. 202001476B0C501, 202000017B0) and waived the informed consent requirement. Patient information de-identified and anonymized before analyses. The authors are responsible for designing, conducting, drafting, and editing this manuscript. Our study complies with the Declaration of Helsinki and the ethical regulations of the CGMH Institutional Review Board.

## 3. Results

### 3.1. Patient Characteristics and Baseline Demographics

We identified 1480 patients with newly diagnosed VTE treated with apixaban (*n* = 265, 18%), dabigatran (*n* = 51, 3.4%), edoxaban (*n* = 151, 10.2%), or rivaroxaban (*n* = 1013, 68.4%) between 1 January 2012, and 31 December 2019 (Figure 1). The distribution of DOACs at different time periods in patients with VTE is shown in Appendix A. The patients treated with rivaroxaban had higher creatinine and D-dimer levels. The patients in the apixaban group were more likely to have diabetes and hypertension. In the edoxaban group, the patients were older and more likely to have cancer. Overall, 51.6% of the patients received standard-dose DOACs, whereas 48.4% received low-dose DOACs. The proportion of patients who received low-dose DOACs was 78.4% in the dabigatran group, 68.8% in the edoxaban group, 55.4% in the apixaban group, and 41.9% in the rivaroxaban group (Table 1). Patients with atrial fibrillation accounted for 7.36% of the total population. No significant intergroup differences in sex, BMI, platelet level, hemoglobin level, liver function, coronary artery disease, atrial fibrillation, Charlson comorbidity index, stroke, liver cirrhosis, myocardial infarction, or chronic obstructive lung disease were note.

### 3.2. Effectiveness and Safety Outcomes

The effectiveness and safety outcomes are listed in Table 2, and the cumulative incidence rates of the outcomes are displayed in Figure 2. Both recurrent VTE and major bleeding occurred in 9.06%, 9.80%, 8.61%, and 10.86% of the patients treated with apixaban, dabigatran, edoxaban, and rivaroxaban, respectively, within 1 year. No significant difference in the composite outcomes was observed between the rivaroxaban group and the apixaban, dabigatran, or edoxaban groups, with the sub-distribution HRs (SHRs) being 0.80, 0.81, and 0.76, respectively. The rate of recurrent VTE was 4.15%, 5.88%, 2.65%, and 3.46% in the apixaban, dabigatran, edoxaban, and rivaroxaban groups, respectively. The rate of major bleeding events was 4.91%, 3.92%, 5.96%, and 7.6% in the apixaban, dabigatran, edoxaban, and rivaroxaban groups, respectively. No significant difference in the safety outcomes of intracranial hemorrhage, major GI bleeding, or all-cause mortality was observed between the rivaroxaban and other DOAC groups (Table 2).

### 3.3. Subgroup Analyses and Risk of Outcomes Among DOACs

The risks of VTE recurrence and major bleeding remained constant across the evaluated DOACs in all prespecified subgroups, except for the impaired renal function (eGFR ≤ 50 mL/min/1.73 m^2^) subgroup. In the impaired renal function subgroup, a lower risk of the composite events was observed among patients taking apixaban than among those taking rivaroxaban. In the analysis of the subgroups, including age ≥ 65 years, pulmonary embolism, hemoglobin level ≤10 g/dL, standard DOAC dose, low DOAC dose, and active cancer status, no significant differences in the composite outcomes were observed between the rivaroxaban and other DOAC groups (Figure 3).

### 3.4. Sensitivity Analysis of Clinical Outcomes in The Stabilized IPTW Cohort

The baseline demographics and clinical characteristics of patients with VTE treated with DOACs after propensity score weighting are shown in Appendix A. The effectiveness and safety outcomes in the stabilized IPTW cohort are listed in Appendix A. No significant difference in the recurrent VTE or major bleeding was observed between the rivaroxaban and other NOAC groups.

## 4. Discussion

To the best of our understanding, this is the first population-based study that evaluated the clinical effectiveness and safety outcomes of four DOACs in Asian patients with VTE. The results revealed that different DOAC treatments were not associated with varying rates of recurrent VTE or major bleeding in the Asian population. Furthermore, in the subgroup analysis, no significant differences in the composite outcomes were observed between the rivaroxaban and other DOAC groups. VTE management is based on balancing the treatment benefits and risk of bleeding; therefore, treatment consideration for Asian patients with VTE is crucial because of the inherited risk of intracranial hemorrhage with vitamin K antagonists. Major phase III clinical trials have suggested that DOACs have advantages over warfarin regarding bleeding outcomes; however, only a few Asian patients were included in these studies [5,6,7,8,9]. Furthermore, data on the direct comparison of effectiveness and safety among different DOACs are limited. Therefore, to overcome this limitation, we specifically focused on the Asian population and provided crucial information on the effectiveness and safety profile of four DOACs.

Approximately half (48.3%) of the patients with VTE were taking low-dose DOACs, thereby indicating that physicians prescribe low-dose DOACs to Asian patients with VTE. This finding is consistent with those of previous studies reporting that lower DOAC doses were more often used in Asian patients with nonvalvular atrial fibrillation [16,17]. The prescription of low-dose DOACs in Asian patients with VTE may be attributed to their relatively small body size and lower renal clearance. In this study, the usage of low-dose DOACs was 78.4%, 68.8%, 55.4%, and 41.9% in the dabigatran, edoxaban, apixaban, and rivaroxaban groups, respectively. Underdosing may be associated with a relatively low risk of bleeding events but may lead to fewer clinical benefits with a higher thromboembolic risk [17]. However, the subgroup analysis demonstrated that in patients treated with low-dose DOACs, no significant differences in the composite outcomes were observed between the rivaroxaban and other DOAC groups. Furthermore, after adjustment for dosage effect, no differences in the composite outcomes (recurrent VTE and major bleeding) were observed among the DOAC groups.

The recurrent VTE rate of the DOACs at 12 months ranged from 2.6% to 5.8%, which was higher than the rates in randomized studies discussing the treatment effect of DOACs. The EINSTEIN DVT and EINSTEIN PE trials included patients with DVT or PE to evaluate and compare the efficacy and safety of rivaroxaban with warfarin [5,6,18]. The recurrent VTE rate of rivaroxaban was 2.1% in the EINSTEIN DVT and PE trials but was 3.5% in our analysis. The recurrent VTE rate of apixaban was 4.2% in our analysis, which was higher than the 2.3% reported in the AMPLIFY study [7]. The recurrent VTE rate of dabigatran was 5.9% in our analysis, which was higher than the 2.3% reported in the pivotal clinical trials [4]. The discrepancy between the recurrent VTE rates in our study and randomized trials may be multifactorial, with the differences in the background characteristics of the study population being a potential explanation. More patients with cancer were noted in our cohort (41.1%) compared with the EINSTEIN program (5.6%), the AMPLIFY study (2.6%), and a trial on dabigatran (3.9%). Studies demonstrated higher rates of recurrent VTE in patients with cancer under warfarin therapy than in patients without cancer [19,20]. A meta-analysis of RCTs, including Re-Cover I and II, EINSTEIN DVT, EINSTEIN PE, and Hokusai-VTE, demonstrated a higher recurrent VTE rate (4.1%) in patients with cancer and receiving DOAC therapy than in patients without cancer (2.6%) [21]. In fact, the previous study demonstrated that Asian patients were more likely to have active cancer (19.8% vs. 8.1%) or a history of cancer (19.1% vs. 12.0%) than patients from the rest of the world [22]. In our previous study comparing DOACs with low-molecular-weight heparin in Asian patients with cancer-associated VTE, the recurrent VTE rate at 1 year in the DOAC group was 7.2% [15]. Therefore, when evaluating the recurrent VTE rate in patients with VTE in real-world practice, the underlying comorbidity of active cancer must be considered because the recurrent VTE rate could be higher in patients with cancer than in those without cancer.

The major bleeding rate among patients treated with different DOACs was higher in our study than in randomized studies evaluating the treatment effects of DOACs. The major bleeding rate of rivaroxaban in our study was 7.6%, which is higher than the 1.0% reported in the EINSTEIN program [18]. Similarly, the incidence of major bleeding in this study was higher than those in RCTs, with the major bleeding rates being 4.9% versus 0.6% for apixaban, 3.9% versus 1.2% for dabigatran, and 5.9% versus 1.4% for edoxaban [4,7,8]. Compared with the study population in a randomized clinical trial, the patients in the current study were older (67.7 vs. 54.7–57.9 years), were more likely to have an eGFR < 50 mL/min/1.73 m^2^ (26.2% vs. 6.5–8.8%), and were more likely to have lower body weight. Older age, impaired renal function, and lower body weight are the risk factors for DOAC-related bleeding events, which may have contributed to the higher major bleeding rate in this study compared with that of a pivotal randomized trial [20]. Furthermore, we included more patients with active cancer compared with an RCT (41.1% vs. 5.1%). Patients with cancer have an increased risk of bleeding, which may be further exacerbated by anticoagulation treatments. Anticoagulant therapy with warfarin is associated with a higher major bleeding rate in patients with cancer-associated VTE than in those with noncancer-associated VTE [19]. A meta-analysis of RCTs reported that the major bleeding rate and clinically relevant nonmajor bleeding rate were higher in patients with active cancer than in those with nonactive cancer (14.8% vs. 7.7%) [21]. Our previous study evaluating DOACs’ safety demonstrated a 7.4% major bleeding rate in cancer associated VTE. Taking together, older age, lower body weight, and a higher proportion of cancer patients in our study may explain the difference in bleeding rates between this study and RCTs.

### Limitations

We used a large-scale database maintained by Taiwan’s largest health care provider, including seven institutions across the country. The study provides essential information on the clinical outcomes of DOAC treatment in patients with VTE; however, this study has some limitations. First, because of the real-world evidence and the retrospective nature of the study, the DOAC groups might have inherent differences. Therefore, we performed adjustments for all the covariates related to the outcomes. The results of major adverse cardiovascular events after adjustment for death as a competing risk remained consistent with those in the primary analyses. Second, we evaluated the 1-year outcome of DOAC treatment in patients with VTE. Therefore, future studies with a longer follow-up are required to confirm the present findings. Third, although we included four DOACs, most patients (68.4%) received rivaroxaban, the first DOAC to be reimbursed by the Taiwan National Health Insurance in 2014 for VTE treatment [23]. The lesser use of other DOACs may have contributed to a potential bias in the event analysis. Therefore, prospective studies are warranted to confirm these results. Fourth, although this is a multi-institutional study in Taiwan, patients in our study may not fully represent populations in other Asian countries. However, the recurrent VTE rate in our study was compatible with the GARFIELD-VTE study, which demonstrated a 5.5% recurrent VTE rate at 12 months for patients from Asia (including China, Hong Kong, Japan, Malaysia, South Korea, Taiwan, and Thailand) [22]. As a final point, noncompliance may lead to the underestimation of the actual use of prescribed drugs, as the information on prescription drugs may not reflect the actual use.

## 5. Conclusions

Compared with RCTs, a higher proportion of patients with active cancer and the frequent use of low-dose DOACs were noted in this study. However, there are more bleeding events and higher recurrent VTE than previously reported in randomized studies. Treatment with different DOACs in Asian patients with VTE was not associated with a significant difference in recurrent VTE or major bleeding. Therefore, in real-world practice, different DOACs have similar effectiveness and safety in the treatment of VTE.

## Figures and Tables

**Figure 1 jcm-11-07159-f001:**
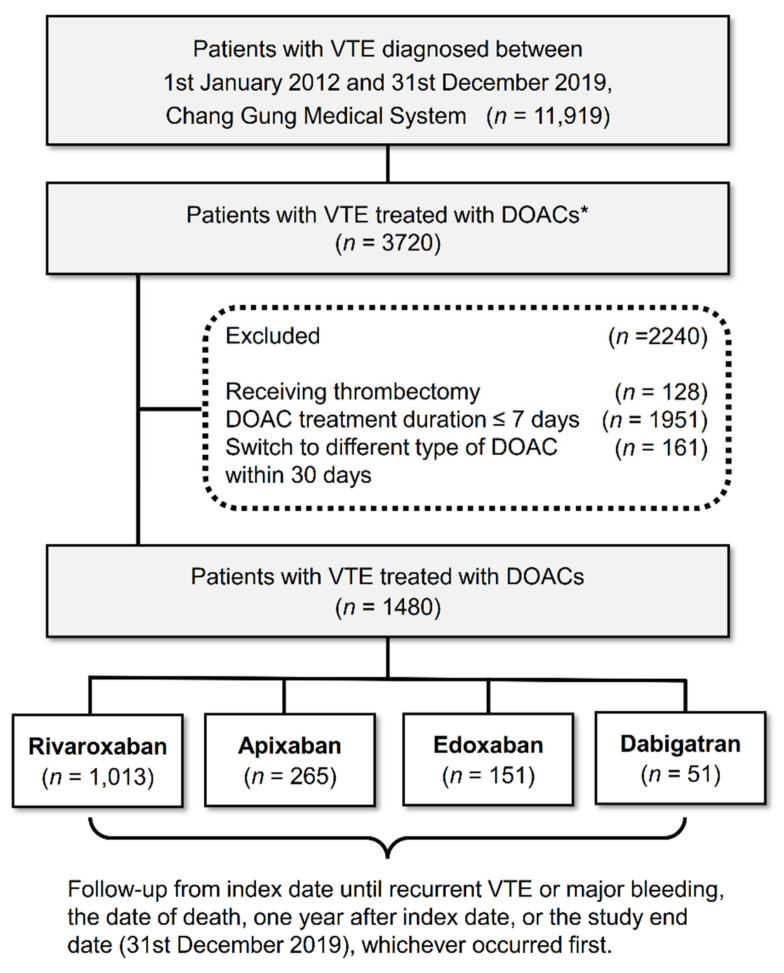
Enrollment and follow-up of study patients. A total of 11,919 patients with newly diagnosed VTE from January 2012 to December 2019 were identified. After relevant exclusion, 1480 patients treated with DOACs were included in the study; of whom, 1013 (68.4%), 265 (18%), 151 (10.2%), and 51 (3.4%) received rivaroxaban, apixaban, edoxaban, and dabigatran, respectively. Those initiated on an anticoagulant therapy for more than 3 months after VTE diagnosis were not included. DOAC, direct oral anticoagulant; VTE, venous thromboembolism. * Those who started anticoagulation therapy more than three months after VTE diagnosis will not be included.

**Figure 2 jcm-11-07159-f002:**
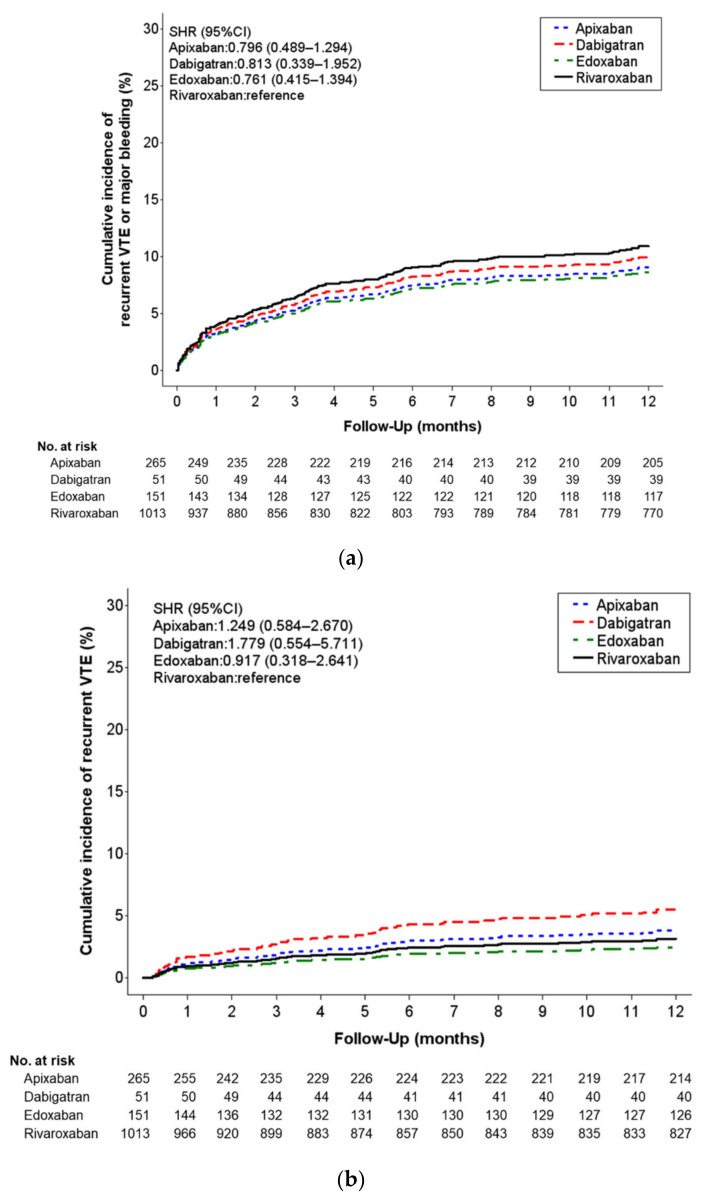
(**a**) Cumulative incidence of recurrent VTE or major bleeding at 1-year follow-up for each DOAC type. (**b**) Cumulative probability of event rates in each study group for recurrent VTE. (**c**) Cumulative probability of event rates in each study group for major bleeding.

**Figure 3 jcm-11-07159-f003:**
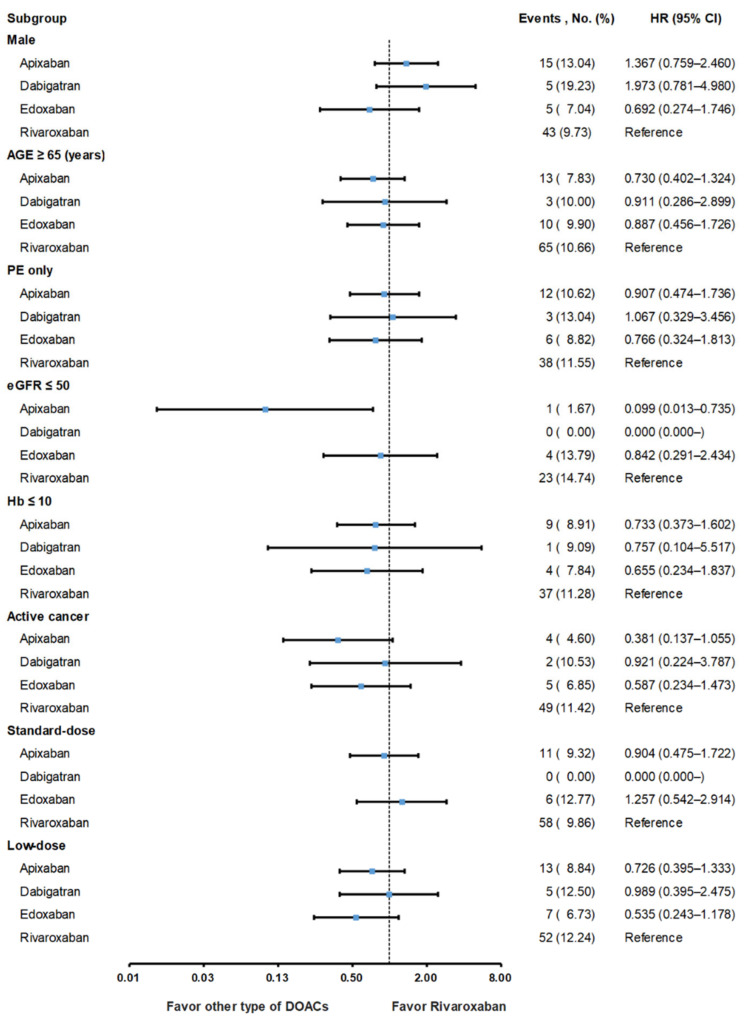
Subgroup analysis for detecting the composite outcomes of recurrent VTE and major bleeding in patients treated with different DOACs. DVT, deep vein thrombosis; eGFR, estimated glomerular filtration rate; Hb, hemoglobin; DOACs, direct oral anticoagulants; PE, pulmonary embolism; VTE, venous thromboembolism.

**Table 1 jcm-11-07159-t001:** Baseline demographics and clinical characteristics of patients with VTE treated with DOACs.

	Total, No. (%) (*n* = 1480)	Apixaban, No. (%) (*n* = 265)	Dabigatran, No. (%) (*n* = 51)	Edoxaban, No. (%) (*n* = 151)	Rivaroxaban, No. (%) (*n* = 1013)	*p* Value
Sex						0.65
Female	826 (55.81)	150 (56.60)	25 (49.02)	80 (52.98)	571 (56.37)	
Male	654 (44.19)	115 (43.40)	26 (50.98)	71 (47.02)	442 (43.63)	
Body weight (kg)	64.02 ± 15.35	64.52 ± 16.93	66.36 ± 18.39	60.52 ± 12.86	64.30 ± 15.05	0.05
BMI	25.90 ± 14.64	26.58 ± 13.91	26.25 ± 6.54	24.28 ± 5.14	25.92 ± 15.94	0.73
Age at index date						
mean$\left(\mathrm{SD}\right)$	67.77 ± 15.89	69.17 ± 16.37	65.25 ± 17.54	70.61 ± 15.93	67.10 ± 15.62	0.02
18–49	223 (15.07)	43 (16.23)	11 (21.57)	16 (10.60)	153 (15.10)	0.03
50–64	350 (23.65)	56 (21.13)	10 (19.61)	34 (22.52)	250 (24.68)	
65–74	335 (22.64)	50 (18.87)	12 (23.53)	30 (19.87)	243 (23.99)	
75–85	391 (26.42)	69 (26.04)	15 (29.41)	46 (30.46)	261 (25.77)	
85+	181 (12.23)	47 (17.74)	3 (5.88)	25 (16.56)	106 (10.46)	
Type of venous thromboembolism						0.004
Pulmonary embolism only	533 (36.01)	113 (42.64)	23 (45.10)	68 (45.03)	329 (32.48)	
Deep-vein thrombosis only	716 (48.38)	115 (43.40)	22 (43.14)	58 (38.41)	521 (51.43)	
PE and DVT	231 (15.61)	37 (13.96)	6 (11.76)	25 (16.56)	163 (16.09)	
Creatinine (mg/dL), mean (SD)	0.95 ± 0.78	1.21 ± 1.37	0.87 ± 0.51	0.90 ± 0.43	0.90 ± 0.58	<0.001
eGFR						<0.001
≤30 mL/min	65 (4.39)	27 (10.19)	2 (3.92)	3 (1.99)	33 (3.26)	
>30 to ≤50 mL/min	323 (21.82)	58 (21.89)	9 (17.65)	39 (25.83)	217 (21.42)	
>50 mL/min	1092 (73.78)	180 (67.92)	40 (78.43)	109 (72.19)	763 (75.32)	
Platelet (1000/μL), mean (SD)	228.91 ± 106.80	235.68 ± 100.72	251.56 ± 107.64	239.46 ± 122.57	224.45 ± 105.60	0.09
Platelet count						0.76
>100,000 per μL	1375 (93.03)	248 (93.58)	49 (98.00)	142 (94.04)	936 (92.49)	
50,000–100,000 per μL	90 (6.09)	16 (6.04)	1 (2.00)	7 (4.64)	66 (6.52)	
<50,000 per μL	13 (0.88)	1 (0.38)	0 (0.00)	2 (1.32)	10 (0.99)	
Hemoglobin (g/dL), mean (SD)	11.17 ± 2.07	10.88 ± 1.95	11.37 ± 1.79	11.21 ± 2.17	11.23 ± 2.10	0.10
Hemoglobin level						0.13
>10 g/dL	989 (66.91)	164 (61.89)	40 (80.00)	100 (66.23)	685 (67.69)	
8–10 g/dL	431 (29.16)	93 (35.09)	9 (18.00)	43 (28.48)	286 (28.26)	
<8 g/dL	56 (3.79)	8 (3.02)	1 (2.00)	8 (5.30)	41 (4.05)	
D-dimer	2896.01 ± 3856.66	2639.40 ± 3820.20	2902.73 ± 3665.89	1627.10 ± 3076.81	3147.15 ± 3943.32	0.001
AST (U/L)	38.51 ± 53.64	40.63 ± 70.05	42.56 ± 75.79	32.01 ± 19.05	38.69 ± 50.83	0.48
ALT (U/L)	32.93 ± 42.26	31.89 ± 55.41	31.24 ± 30.23	29.15 ± 24.86	33.86 ± 46.79	0.66
Charlson comorbidity index						0.05
0	259 (17.5)	42 (15.85)	10 (19.61)	17 (11.26)	190 (18.76)	
1–2	416 (28.11)	68 (25.66)	13 (25.49)	36 (23.84)	299 (29.52)	
3+	805 (54.39)	155 (58.49)	28 (54.90)	98 (64.90)	524 (51.73)	
Comorbidities						
Active cancer ^a^	608 (41.08)	87 (32.83)	19 (37.25)	73 (48.34)	429 (42.35)	0.008
Myocardial infarction	38 (2.57)	7 (2.64)	2 (3.92)	3 (1.99)	26 (2.57)	0.85
Stroke	223 (15.07)	52 (19.62)	8 (15.69)	21 (13.91)	142 (14.02)	0.15
Coronary artery disease	208 (14.05)	47 (17.74)	7 (13.73)	28 (18.54)	126 (12.44)	0.05
Diabetes mellitus	395 (26.69)	85 (32.08)	13 (25.49)	48 (31.79)	249 (24.58)	0.04
Hypertension	802 (54.19)	173 (65.28)	26 (50.98)	86 (56.95)	517 (51.04)	0.001
Liver cirrhosis	56 (3.78)	15 (5.66)	1 (1.96)	9 (5.96)	31 (3.06)	0.09
Chronic obstructive lung disease	239 (16.15)	49 (18.49)	6 (11.76)	29 (19.21)	155 (15.30)	0.341
Atrial fibrillation	109 (7.36)	26 (9.81)	5 (9.80)	16 (10.60)	62 (6.12)	0.06
Doses of DOACs						<0.001
Standard-dose	764 (51.62%)	118 (44.53%)	11 (21.57%)	47 (31.13%)	588 (58.05%)	
Low-dose	716 (48.38%)	147 (55.47%)	40 (78.43%)	104 (68.87%)	425 (41.95%)	
Drug prescribed within 30 days of index date						
Aspirin	137 (9.26)	31 (11.7)	4 (7.84)	9 (5.96)	93 (9.18)	0.27
P2Y12 inhibitor ^b^	77 (5.20)	18 (6.79)	3 (5.88)	9 (5.96)	47 (4.64)	0.52
NSAIDs	306 (20.68)	48 (18.11)	10 (19.61)	29 (19.21)	219 (21.62)	0.61

ALT, Alanine aminotransferase; AST, Aspartate Transaminase; BMI, body mass index; DOACs, direct oral anticoagulants; DVT, deep vein thrombosis; eGFR, estimated glomerular filtration rate; NSAIDs, non-steroidal anti-inflammatory drugs; PE, pulmonary embolism; VTE, venous thromboembolism. ^a^ Cancer diagnosis within 6 months of the index date, metastatic cancer, hematology cancer, patients with cancer treated with radiotherapy or systemic therapy, patients with recurrent cancer, or patients with more than two oncology outpatient visits within 1 year. ^b^ clopidogrel or ticagrelor.

**Table 2 jcm-11-07159-t002:** Effectiveness and safety clinical outcomes.

	VTE Treated with DOAC ^a^					HR or SHR for Apixaban (95% CI) ^b^	HR or SHR for Dabigatran (95% CI) ^b^	HR or SHR for Edoxaban (95% CI) ^b^
	Total (*n* = 1480)	Apixaban (*n* = 265)	Dabigatran (*n* = 51)	Edoxaban (*n* = 151)	Rivaroxaban (*n* = 1013)			
Composite effectiveness and safety outcome Recurrent VTE or major bleeding	152 (10.27)	24 (9.06)	5 (9.80)	13 (8.61)	110 (10.86)	0.80 (0.49–1.29)	0.81 (0.34–1.95)	0.76 (0.42–1.39)
Effectiveness outcome								
Recurrent VTE	53 (3.58)	11 (4.15)	3 (5.88)	4 (2.65)	35 (3.46)	1.25 (0.58–2.67)	1.78 (0.55–5.71)	0.92 (0.32–2.64)
Recurrent DVT	28 (1.89)	6 (2.26)	0 (0.00)	1 (0.66)	21 (2.07)	1.40 (0.49–4.00)	NA	0.57 (0.06–5.66)
Recurrent PE	26 (1.76)	5 (1.89)	3 (5.88)	3 (1.99)	15 (1.48)	1.23 (0.46–3.27)	3.28 (0.73–14.64)	1.16 (0.32–4.15)
Safety outcome								
Major bleeding ^c^	101 (6.82)	13 (4.91)	2 (3.92)	9 (5.96)	77 (7.60)	0.59 (0.31–1.13)	0.47 (0.11–1.97)	0.70 (0.34–1.46)
Intra-cranial hemorrhage	26 (1.76)	7 (2.64)	0 (0.00)	0 (0.00)	19 (1.88)	1.12 (0.44–2.81)	NA	NA
Major GI bleeding	17 (1.15)	0 (0.00)	1 (1.96)	0 (0.00)	16 (1.58)	NA	1.05 (0.10–11.06)	NA
Other critical site bleedings	53 (3.58)	5 (1.89)	1 (1.96)	8 (5.30)	39 (3.85)	0.44 (0.16–1.23)	0.58 (0.08–4.49)	1.37 (0.60–3.13)
Decrease in Hb of ≥2 g/dL	9 (0.61)	2 (0.75)	0 (0)	1 (0.66)	6 (0.59)	3.53 (0.91–13.74)	NA	0.92 (0.02–43.50)
Death from any cause	190 (12.84)	34 (12.83)	5 (9.80)	16 (10.60)	135 (13.33)	1.02 (0.69–1.52)	0.79 (0.32–1.97)	0.68 (0.40–1.15)

DOAC, direct oral anticoagulant; DVT, deep vein thrombosis; HR, hazard ratio; NA, not applicable; PE, pulmonary embolism; SHR, sub-distribution hazard ratio; VTE, venous thromboembolism. ^a^ Values are presented as n (%). ^b^ With adjustments for all variables listed in Table 1. ^c^ Major bleeding included major gastrointestinal bleeding, intracranial hemorrhage, bleeding at other critical sites, and a hemoglobin decrease of 2 g/dL or more over 24 h. For patients who had more than one event, only the first event was counted.

## Data Availability

The original contributions presented in the study are included in the article and supplementary material; further inquiries can be directed to the corresponding author.

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
