# Peer review of "A Comparison among Nonvitamin K Antagonist Oral Anticoagulants in Asian Patients with Venous Thromboembolism: A Multi-Institutional Study"

_jcm, 2022, doi:10.3390/jcm11237159_

Round 1
Reviewer 1 Report
Review JCM-1980898
The manuscript is well written and provides a comprehensive summary from real-world observational data of the relation between clinical outcome and oral anticoagulation therapy in Asian patients with deep vein thrombosis or pulmonary emboli. The text discusses the available data but also points out fields where more research is needed. I believe that the manuscript is suitable for publication with minor revisions.
The article of Ming-Lung Tsai presents a retrospective analysis of a multi-center cohort study between 2012 and 2019 on Asian patients with VTE who received oral anticoagulation with dabigatran, rivaroxaban, apixaban or edoxaban. The main finding is that treatment with different DOACs in Asian patients with VTE was not associated with a significant difference in recurrent adverse events within 1-year follow-up.
Options for anticoagulation have been expanding steadily over the past few decades, providing a greater number of agents for prevention and management of thromboembolic disease. In addition to heparins and vitamin K antagonists, anticoagulants that directly target the enzymatic activity of thrombin and factor Xa have been developed.
Although the incidence in the Asian population is lower than that in the Western population, the burden of VTE in the Asian population is expected to increase with the rapid aging and increased life expectancy of this population.
Abstract: The length of the abstract is 221 words.
Strengths: As for originality it is an original study with data reporting from a large database on VTE. Concerning significance, the study focuses on an important problem of current interest. Appropriate information was provided on the statistical methods. The number of background references was appropriate with important value. The cited references are mostly recent publications without an excessive number of self-citations.
Figures: the charts in the pdf are fair quality and easy to interpret.
The English of the text is of good quality. Minor grammatical mistakes should be corrected.
Study setting: remove unnecessary repetitions
The main limitation is the predominant use of rivaroxaban compared to the other 3 DOACs which might have contributed to selection bias in the event analysis. The manuscript is well-written and provides rigorous and comprehensive research, however, the statistical methodology raises some minor issues.
Minor comments:
Q1. January 1, 2012, and January 31, 2019, (line 71) please correct the dates to 1st January 2012 and 31st January 2019 and in Figure 1.
Q2. The term “novel” was initially applied to dabigatran (Pradaxa) when it was introduced to the US market in 2010, followed by introducing factor Xa inhibitors. After 12 years, the idea of NOACs being novel does not seem applicable. Therefore, please replace it with DOAC, direct oral anticoagulant.
Q3. Venous thromboembolism and atrial fibrillation (AF) implicate a high burden of mortality and morbidity. Were data on atrial fibrillation patients collected?
Q4. The high number of rivaroxaban use seemed very interesting. Can the authors explain the preferred treatment choice of rivaroxaban? Has the reimbursement system changed since 2014?
Q5. The authors state that data were extracted on the use of medications, including aspirin, P2Y12 inhibitors, and nonsteroidal anti-inflammatory drugs, 30 days before NOAC initiation were extracted. It is known that clopidogrel use can be associated with high residual platelet reactivity which can result in an increased risk of adverse ischemic events. On the other hand, Asian population has a higher possibility of developing low platelet reactivity resulting in increased bleeding risk. Was any statistical analysis made on the type and duration of P2Y12 inhibitors?
Q6. Were liver function tests and adherence (quantified by capsule counts) routinely assessed?
Q7. Please revise the reference list and correct citations. Reference 14. Volume, Issue and page are missing. DOI number should be given if applicable.
Major comments
Q1. Because of the nonrandomized nature of the study, propensity score analysis should have been performed to minimize any selection bias caused by differences in the clinical characteristics between groups.
Author Response
Point 1: January 1, 2012, and January 31, 2019, (line 71) please correct the dates to 1st January 2012 and 31st January 2019 and in Figure 1.
Response 1: We have accordingly corrected the dates in the paragraph and in Figure 1.
Point 2: The term “novel” was initially applied to dabigatran (Pradaxa) when it was introduced to the US market in 2010, followed by introducing factor Xa inhibitors. After 12 years, the idea of NOACs being novel does not seem applicable. Therefore, please replace it with DOAC, direct oral anticoagulant.
Response 2: We have accordingly replaced the NOACs with DOACs in the paragraph.
Point 3: Venous thromboembolism and atrial fibrillation (AF) implicate a high burden of mortality and morbidity. Were data on atrial fibrillation patients collected?
Response 3: Thank you for the comments. Patients with atrial fibrillation account for 7.36 % of the total population. We added the atrial fibrillation data to Table 1 and modified the text in the “RESULTS” (line 156). “Patients with atrial fibrillation account for 7.36 % of the total population. No significant intergroup difference in sex, BMI, platelet level, hemoglobin level, coronary artery disease, atrial fibrillation, …, and chronic obstructive lung disease was noted.”
Point 4: The high number of rivaroxaban use seemed very interesting. Can the authors explain the preferred treatment choice of rivaroxaban? Has the reimbursement system changed since 2014?
Response 4: Thank you for the comments. The treatment choice of rivaroxaban was according to the physician's decision-making. However, as we mentioned in the “Limitation” (line 277), rivaroxaban was the first NOAC to be reimbursed for venous thromboembolism by the Taiwan National Health Insurance. We have further analyzed the distribution of DOACs at different periods in patients with VTE and provided in eFigure 1 in the supplement (mentioned in manuscript, line 148). After 2016, the rate of rivaroxaban use decreased gradually, whereas that of edoxaban, dabigatran, and apixaban increased progressively.
Point 5: The authors state that data were extracted on the use of medications, including aspirin, P2Y12 inhibitors, and nonsteroidal anti-inflammatory drugs, 30 days before NOAC initiation were extracted. It is known that clopidogrel use can be associated with high residual platelet reactivity which can result in an increased risk of adverse ischemic events. On the other hand, Asian population has a higher possibility of developing low platelet reactivity resulting in increased bleeding risk. Was any statistical analysis made on the type and duration of P2Y12 inhibitors?
Response 5: Thank you for the comment. Unfortunately, only 77 (5.2%) patients in our study received P2Y12 inhibitors. Of them, 73 received clopidogrel, 6 received ticagrelor, and none received prasugrel. Due to the limited number of patients receiving P2Y12 inhibitors, we could not make a statistical analysis of the type and duration of P2Y12 inhibitors.
Point 6: Were liver function tests and adherence (quantified by capsule counts) routinely assessed?
Response 6: Thank you for the comments. We have analyzed the study population's liver function (including AST and ALT) and added it to table 1. We have addressed this point in the “Results” (ling 157). “No significant intergroup difference in sex, BMI, …, liver function, …, and chronic obstructive lung disease was noted. Liver function test was routinely assessed in the CGMH system. The liver function assessment frequency is provided in the table below. As for adherence, due to the limitation of CGRD, we could not evaluate the adherence of the prescription drugs. We have previously addressed this point in the “Limitation” (line 286). “As a final point, noncompliance may lead to underestimation of the actual use of prescribed drugs, as the information on prescription drugs may not reflect the actual use.”
Table. Liver function evaluation frequency of the study population
|
frequency |
Total (n=1480) |
Apixaban (n=265) |
Dabigatran (n=51) |
Edoxaban (n=151) |
Rivaroxaban (n=1013) |
|
Times per year, mean ± SD |
5.59 ± 6.46 |
5.53 ± 6.69 |
5.08 ± 4.27 |
5.56 ± 5.98 |
5.64 ± 6.48 |
Point 7: Please revise the reference list and correct citations. Reference 14. Volume, Issue, and page are missing. DOI number should be given if applicable.
Response 7: We have revised the reference list and corrected the citations accordingly (line 345).
Point 8: Because of the nonrandomized nature of the study, propensity score analysis should have been performed to minimize any selection bias caused by differences in the clinical characteristics between groups.
Response 8: Thank you for the suggestions. We have further performed the propensity score analysis as the sensitivity analysis and addressed it in the “Statistical analysis” (line 128). “Finally, we created a stabilized inverse probability of treatment weighting (IPTW) cohort based on the propensity score, which was calculated using the values of the covariates listed in eTable 3 in the supplement. The risks of recurrent VTE or major bleeding between the groups were compared using a Fine and Gray subdistribution hazard model that considered death as a competing risk. To further rule out possible residual confounding even after IPTW, we adjusted for the covariates with an absolute STD value >0.1 in the survival models.” We also added the paragraph in the “Results” (line 188-190). “The effectiveness and safety outcomes in the IPTW cohort are listed in eTable 4 in the supplement. No significant difference in the recurrent VTE or major bleeding was observed between the rivaroxaban and other NOAC groups.”

Reviewer 2 Report
Current study is a retrospective comparative analysis of 4 different NOACS in the treatment of VTE and their associated rates of bleeding in a Taiwanese population.
Overall, this a good study and provides an important addition to the literature.
Major Points:
It is definitely troubling to see a higher rate of recurrent VTE and a much higher rate of bleeding in this patient population compared to randomized data despite the fact that around half of the patients were on low dose AC.
Seems like cancer status is a major part of your argument regarding higher rates. Can you provide data including recurrent VTE and recurrent bleeding in non-cancer patients separately?
Can you compare your rates of recurrent VTE and major bleeding to other “Asian cohort” evaluating similar outcomes if any.
Minor:
First two lines in the Abstract: “The clinical effectiveness and safety of different nonvitamin K antagonist 19 oral anticoagulants (NOACs) in patients with venous thromboembolism (VTE) remains unclear”
This is just not true, clinical effectiveness and safety of different NOACS in VTE is very well established with numerous randomized controlled trials showing that. Based on your paper, what you probably tried to say there are no comparative studies between different NOACs, especially in the Asian population, please correct and clarify that.
Line 40:
“Death occurs in approximately 6% of patients with DVT and 12% of those with PE without 40 proper treatment [2]” Is this per year?? Specify.
Figure 2: I would specify on each graph what is the outcome rather than just in the figure legend
In the limitations I would clarify that your population does not really represent the “Asian population” but rather a Taiwanese population which is more homogeneous.
I would emphasize in the conclusion, that one of the major finding in your study is that the selected Asian population bleeds more and has higher recurrent VTE than previously reported in trials.
Author Response
Point 1: It is definitely troubling to see a higher rate of recurrent VTE and a much higher rate of bleeding in this patient population compared to randomized data despite the fact that around half of the patients were on low-dose AC. It seems like cancer status is a major part of your argument regarding higher rates. Can you provide data including recurrent VTE and recurrent bleeding in non-cancer patients separately?
Response 1: Thank you for the comment. The effectiveness and safety outcomes in patients with and without cancer are provided in the table below. Compared to the patients without active cancer, patients in the cancer group had a higher risk of recurrent VTE (5.59% vs. 2.17%), major bleeding (7.40% vs. 6.42%), major GI bleeding (1.64% vs. 0.8%), and composite VTE and major bleeding (12.66% vs. 8.6%). We have also added one sentence in the “Discussion” (line 237). “…the previous study demonstrated that Asian patients were more likely to have active cancer (19.8% vs. 8.1%) or a history of cancer (19.1% vs. 12.0%) than patients from the rest of the world.”
Table. Effectiveness and safety clinical outcomes of cancer and non-cancer population
|
|
Non-cancer population (n=872) |
Cancer population (n=608) |
|
Recurrent VTE or major bleeding |
75 (8.6) |
77 (12.66) |
|
Recurrent VTE |
19 (2.17) |
34 (5.59) |
|
Major bleeding |
56 (6.42) |
45 (7.40) |
|
Major GI bleeding |
7 (0.80) |
10 (1.64) |
|
Death from any cause |
46 (5.28) |
144 (23.68) |
Point 2: Can you compare your rates of recurrent VTE and major bleeding to other “Asian cohort” evaluating similar outcomes if any.
Response 2: Thank you for the comments. We have added the paragraph in the “Limitations” (line 283). “However, the recurrent VTE rate in our study was compatible with the GARFIELD-VTE study, which demonstrated a 5.5% recurrent VTE rate at 12 months for patients from Asia (including China, Hong Kong, Japan, Malaysia, South Korea, Taiwan, and Thailand).”
Point 3: First two lines in the Abstract: “The clinical effectiveness and safety of different nonvitamin K antagonist oral anticoagulants (NOACs) in patients with venous thromboembolism (VTE) remains unclear” This is just not true, clinical effectiveness and safety of different NOACS in VTE is very well established with numerous randomized controlled trials showing that. Based on your paper, what you probably tried to say there are no comparative studies between different NOACs, especially in the Asian population, please correct and clarify that.
Response 3: Thank you for the comment. We have modified the test in the Abstract (line 19). “Comparison of clinical effectiveness and safety of across different nonvitamin K antagonist oral anticoagulants (NOACs) in Asian patients with venous thromboembolism (VTE) remains unclear.”
Point 4: Line 40: “Death within one year occurs in approximately 6% of patients with DVT and 12% of those with PE without proper treatment [2]” Is this per year?? Specify.
Response 4: We have corrected the typo and modified the text to “Death within one year occurs in approximately 14.6% of patients with DVT and 52.3% of those with PE without proper treatment” (line 40)
Point 5: Figure 2: I would specify on each graph what is the outcome rather than just in the figure legend
Response 5: Thank you for the suggestion. We have added the outcome on each graph in figure 2.
Point 6: In the limitations, I would clarify that your population does not really represent the “Asian population” but rather a Taiwanese population which is more homogeneous.
Response 6: Thank you for the comments. We have added the paragraph in the “Limitations” (line 281). “Fourth, although this is a multi-institutional study in Taiwan, patients in our study may not fully represent populations in other Asian countries.”
Point 7: I would emphasize in the conclusion, that one of the major finding in your study is that the selected Asian population bleeds more and has higher recurrent VTE than previously reported in trials.
Response 7: Thank you for the comments. We have added the sentence in the “CONCLUSION” (line 291-292). “However, there are more bleeding events and higher recurrent VTE than previously reported in randomized studies.”

Round 2
Reviewer 2 Report
Authors addressed all the comments